# Bone Healing Evaluation Following Different Osteotomic Techniques in Animal Models: A Suitable Method for Clinical Insights

**Alexandre Anesi** [1,*], **Mattia Di Bartolomeo** [2], **Arrigo Pellacani** [2], **Marzia Ferretti** [3], **Francesco Cavani** [3], **Roberta Salvatori** [1], **Riccardo Nocini** [4], **Carla Palumbo** [3] **and Luigi Chiarini** [1]

1   Department of Medical and Surgical Sciences for Children & Adults, Cranio-Maxillo-Facial Surgery, University of Modena and Reggio Emilia, Largo del Pozzo 71, 41124 Modena, Italy; roberta.salvatori@unimore.it (R.S.); luigi.chiarini@unimore.it (L.C.)

2   Unit of Dentistry and Maxillo-Facial Surgery, Surgery, Dentistry, Maternity and Infant Department, University of Verona, P.le L.A. Scuro 10, 37134 Verona, Italy; mattiadiba@hotmail.it (M.D.B.); arrigo.pellacani@libero.it (A.P.)

3   Section of Human Morphology, Department of Biomedical, Metabolic and Neural Sciences, University of Modena and Reggio Emilia, Largo del Pozzo 71, 41125 Modena, Italy; marzia.ferretti@unimore.it (M.F.); francesco.cavani@unimore.it (F.C.); carla.palumbo@unimore.it (C.P.)

4   Section of Ear Nose and Throat (ENT), Department of Surgical Sciences, Dentistry, Gynecology and Pediatrics, University of Verona, 37124 Verona, Italy; riccardo.nocini@gmail.com

*   Correspondence: alexandre.anesi@unimore.it; Tel.: +39-059-422-4552

**Abstract:** Osteotomy is a common step in oncological, reconstructive, and trauma surgery. Drilling and elevated temperature during osteotomy produce thermal osteonecrosis. Heat and associated mechanical damage during osteotomy can impair bone healing, with consequent failure of fracture fixation or dental implants. Several ex vivo studies on animal bone were recently focused on heating production during osteotomy with conventional drill and piezoelectric devices, particularly in endosseous dental implant sites. The current literature on bone drilling and osteotomic surface analysis is here reviewed and the dynamics of bone healing after osteotomy with traditional and piezoelectric devices are discussed. Moreover, the methodologies involved in the experimental osteotomy and clinical studies are compared, focusing on ex vivo and in vivo findings.

**Keywords:** bone healing; bone damage; bone injury; bone surgery; osteotomy; drilling; piezosurgery; in vivo; ex vivo

## 1. Historical Background

Osteotomy consists of cutting the bone. In clinical practice, it is a surgical procedure in which a bone is divided or a piece of bone is excised.

Specifically, bone cutting techniques can be classified into two types: osteotomy that requires a full-thickness division of the bone and corticotomy in which only the bone cortex is divided with preservation of the periosteal and endosteal layers [1].

Osteotomy is an ancient surgical skill dating back to Hippocrates circa 415 BC, who used a new fracture to improve the alignment of a previously angulated humerus because of trauma [2].

Corticotomy was first described by Ilizarov in the 1950s as a low-energy division of the bone cortex preserving both the periosteum and the medullary vascularization in order to increase bone neoformation and elongation [3] (hereinafter referred to as distraction osteogenesis). Since then, several

osteotomy procedures have been developed for deformity correction or bone lengthening—in particular, by De Bastiani et al. during the 1980s [2].

In 1991, Paly and Tetsworth introduced the percutaneous Gigli saw technique, also known as the Afghan technique, as a new procedure for performing minimally invasive low-energy osteotomies in long bones that leaves a very smooth cut, especially important for rotational correction [2,4,5]. Shortened duration of surgery, minimal soft-tissue damage and periosteum preservation make this technique one of the most recommended still today [4,6,7]. According to our experience, however, a low level of consistency between planning and execution characterizes the geometry of the osteotomy plane by Gigli saw, because of the intrinsic poor control of the device with complex angulations.

Later, motorized rotatory and oscillating cutting tools were developed, which have greatly simplified the surgical performance, allowing a precise and direct cut. Nevertheless, power saw osteotomies have several disadvantages, such as the requirement for a relatively large open exposure, the risk of tissue overheating resulting in possible thermal necrosis and soft tissue trauma.

In order to reduce these complications, ultrasound-based osteotomy techniques (Piezosurgery®) have been developed in recent years [8]. In particular, its development was encouraged by the need for high levels of precision and safety in bone surgery compared with that achieved by standard bur and saw instruments [9].

Parallel to the latter method, interest in the possibility of applying lasers to surgical osteotomies has also grown in recent years. In general, several studies have shown that the advantages of laser osteotomy are numerous, such as sterilization of the operative field (bactericidal effect) and simultaneous hemostasis during the cut, resulting in less bleeding and risk of infections, less post-operative pain and edema, faster recovery, need for smaller surgical access and limited scarring [10]. The Er:YAG laser, in particular, would seem to minimize the thermal damage of the bone and surrounding tissues (5 μm) [11] and promote faster healing due to the osteoinductive effect, as well as allowing a very precise cut and minimal invasiveness [12,13].

## 2. Osteotomy in Clinical Practice

According to Giraud et al. [14], osteotomies can be classified based on four criteria:

- End result
- Type
- Approach
- Incision site

The "end result" means the reason for performing the osteotomy, which can be specified as follows: (a) excision osteotomy for removal of a pathological piece of bone; (b) reparative osteotomy for anatomical correction and (c) osteotomy for approaching purpose, which consists in cutting a bone that obstructs access to the main operation site (for example, the tumor site) and then repositioning it in its anatomical location.

The "type" of osteotomy corresponds to the figure of the cut, for each of which a specific tool is designed.

Subperiosteal and extraperiosteal are the possible "approaches" to osteotomy, whereas the "incision site" localizes the bone region cut in long bones (epiphyseal, metaphyseal or diaphyseal) or in short/flat/irregular bones (carpus, skull, vertebrae, facial bones, et al.).

The goal of osteotomy surgery varies according to the surgical field considered.

Osteotomy or bone cutting is a widely standardized surgical procedure used for multiple purposes such as allowing access to malignancies, removing pathological bone fragments or realigning deformed bone segments.

Similarly, bone drilling is a complementary surgical technique mainly used in fracture fixation, allowing screw insertion [15]; moreover, it represents an essential step in joint [16] and oral [17] implantology.

In orthopedic surgical correction of skeletal anomalies, osteotomies are performed broadly for two purposes: a simple osteotomy to acutely realign the axis of the bone and those techniques which allow bone lengthening or bone transport [2]. In orthopedic surgery, a short healing time is particularly important to avoid functional long-term complications (malconsolidation) [18].

In oncological and reconstructive surgery, osteotomy is commonly a mandatory step to access and remove bone or extraosseous malignancies as well as to grant both functional and aesthetic satisfactory reconstruction outcomes.

Most of the head and neck carcinomas involving the jaws require surgical osteotomy followed by adjuvant radiotherapy or concomitant chemoradiotherapy [19]. The effects of radiation on bone are well known and include alterations in the mineralized structure, alterations in the biomechanics of the collagen structure, a decrease in the number and qualitative osteogenic cell activity and cytokine alterations resulting in bone remodeling delay and damage [19]. However, few studies have focused on primary reconstruction followed by early irradiation but many of the currently available experimental models seem to be altered by radiation [19]. In the surgical reconstruction of mandibular bone defects with biomaterials, the early administration of radiation would seem to favor reabsorbing phenomena [20]. Similar experimental results on animal models have been found in the field of dental implantology [21,22]. Schon et al. [23] observed both significant quantitative and qualitative bone alterations with delayed initiation of bone formation after irradiation.

To guarantee cancer patients to receive adjuvant radiotherapy within the recommended times, it is important to achieve a good degree of bone healing in a short time after oncological and reconstructive surgery. In fact, considering that the normal time for secondary bone callus formation is around 4–6 weeks [24] and adjuvant radiotherapy should be ideally started within 42 days after surgery, it is easy to understand how a delay in bone healing implies a worse oncological prognosis [25].

## 3. Osteotomy-Related Tissue Damage

All surgical practices potentially expose the bone and surrounding tissues to damage. In particular, the tissue insult is mainly due to the heat produced by the friction generated between the osteotome saw or the drill bit and the bone [15].

It has been well established that, if the temperature exceeds the universally accepted danger threshold of 47 °C for 1 min or more, thermal necrosis can be produced [15,16,26–28]. As the temperature and/or duration of the cutting procedure increases, the severity of the damage improves up to irreversibility [15,28].

Thermal osteonecrosis substantially consists in bone cell death and subversion of endosteal architecture [27]. Specifically, immediate (within a few minutes) and delayed (within a few hours) effects of thermal shock can be recognized: the former consists of swelling and destruction of cell membranes and coagulation, with relative loss of blood flow leading to cell necrosis, whereas the latter refers to the activation of cellular signaling pathways that lead to apoptosis [27,28]. In both cases, an inflammatory response is always associated. The direct consequence of these events is the resorption of the bone around the screws or implants and its replacement with fibrous connective tissue [28].

Regarding the factors that influence the extent of heat production, there is currently widespread consensus that the main parameter is rotational speed; therefore, a higher speed number (≥3000 rpm) results in higher damage in terms of osteonecrosis [15]. Also relevant in this field are the drilling depth, the bone density and cortical thickness [29], the diameter of the drill bit and the pressure applied [30] and finally the one-step or graduated method of drilling [31,32].

Thermal osteonecrosis largely influences the patient's postoperative outcome, possibly causing screw-mediated bone fixation failure, nonunion or malunion, infections [27], implant loss and delayed healing [33].

Reasonably, cooling during surgical osteotomies or drilling procedures appears mandatory to avoid these complications. In this context, external irrigation with saline is the most common method

adopted since Matthews and Hirsch have broadly demonstrated that it is the most effective cooling method of limiting excessive thermal increase [30].

To date, histology has represented the gold standard for the study (Table 1) of heat-related bone tissue necrosis since it allows an in situ analysis of the cells composing it [34]. Histomorphologically, the assessment of the extent of osteonecrosis is based on the osteocyte condition in the bone lacunae [15,35]; in fact, the death of these cells in the pathological bone is documented by the empty appearance of the lacunae [35]. Bentolila et al. found that osteocyte damage develops through four stages: (1) normal osteocyte, (2) shrunken osteocyte, (3) osteocyte with pyknotic nucleus and (4) empty lacunae [36].

**Table 1.** Main investigations for heat-related bone tissue necrosis.

| Cytology and Histology | Molecular Biology and Biochemistry | Physics |
|---|---|---|
| Histologic parameters<br>• cell composition<br>• blood clot and vessels<br>• connective tissue, bone neoformation, microcracks, lacunae, bridges | PCR<br>BMP, Wnt, osteogenesis, inflammatory cytokines, apoptosis, growth-factors; Hsp70 | Temperature rise<br>• thermocouples<br>• infrared thermography<br>• thermosensors<br>• fiber optic thermometer |
| Histomorphometric and Micromorphometric Analysis | ELISA<br>test on the perisulcular fluid to search for RANKL and OPG levels | Micro-CT |
| Immunohistochemistry<br>• VEGF, CAS3, OPG, RANKL, OC, NADH2 and NADPH2, diaphorase activity | | CBCT |
| Scanning electron microscopy (SEM) | | Laser 3D scanning |
| Transmission electron microscopy (TEM) | | Laser profilometry |
| Intravital microscopy | | |

BMP: Bone Morphogenetic Protein.

However, it has recently been shown that histological/histomorphological investigation techniques do not allow us to grasp the micro-mechanical damage induced by bone cutting surgeries within the drilled sub-surface. Using micro-mechanical investigations such as compression and tensile tests and microhardness, it has been revealed that micro-mechanical bone damage assessment is particularly important [37]. It is plausible that high temperatures during surgical cutting or drilling imply greater damage also in terms of elastic and plastic property alterations of the surrounding non-necrotic bone tissue.

Considering that the bone remodeling process closely depends on the applied functional loads [38], the lower mechanical resistance together with the thermal necrosis could have significant repercussions for the post-surgical healing of osteotomies.

Finally, it is necessary to underline that micro-crack (micro-fractures) formation in the adjacent bone to the cutting or drilling site constitutes another possible complication to consider [39] in clinical practice. The consequence can be prosthetic or bone failure. This risk would appear to be greater if blunt/worn tips are used [16,40].

## 4. How Bone Healing Occurs after Bone Injury

Bone healing is the process that restores the anatomy and function of bone after injury (fracture or osteotomy); it can be divided into primary and secondary healing based on differences in the mobility between the fracture fragments [41,42].

Primary healing happens when bone injured surfaces are juxtaposed and fixed through surgery and bone remodeling through the original fracture line leads to bone healing. Secondary healing takes place in all other circumstances and is usually divided into four stages that partially overlap each other.

The first stage is called the "inflammatory stage": after injury, bone blood vessels release blood within the fracture site and hematoma develops between bone fragments. This lasts a few days and is characterized by pain and swelling. The lack of blood supply to adjacent bone leads to bone necrosis that is characterized histologically by empty osteocyte lacunae. After blood clot formation, cytokines (such as PDGF, TGF-β1, VEGF, PGE1 and E2) released from platelets and mast cells stimulate neoangiogenesis and the formation of a granulation tissue that replaces the blood clot. Mononuclear phagocytes derived from new vessels assist the removal of necrotic bone and blood clots and aid the construction of the soft callus that will follow. Macrophages are also believed to play a fundamental role in fracture repair since they secrete several growth factors, such as fibroblast growth factor (FGF) that initiates fibroplasia both in soft tissue as well as in bone repair [43].

The second stage is called the "soft callus stage": it is characterized by a fibrous tissue whose cells are derived from endosteum, periosteum, bone marrow and adjacent soft tissues. From this stage on, a recapitulation of bone histogenesis occurs, leading to the third stage, called the "hard callus stage", whose progression strictly depends on the presence of blood supply.

If a good blood supply is present, new osteoblasts differentiate and start to lay down the bone matrix during "intramembranous ossification". During intramembranous ossification, both in physiological organogenesis and in pathologic conditions, it has been demonstrated for the first time [8,44–47] that two different processes of bone formation exist, occurring in sequence, named static osteogenesis (SO) and dynamic osteogenesis (DO). SO is characterized by pluristratified cords of "stationary" osteoblasts which differentiate by inductive stimuli [48–51] at roughly constant distance from the capillaries (without moving during their transformation into osteocytes from the differentiation site); otherwise, DO is performed by the typical monostratified laminae of "movable" osteoblasts. The following events occur in sequence: firstly, variously polarized stationary osteoblasts (irregularly arranged inside cords) give rise, in the same place where they differentiate, to osteocytes (clustered within confluent lacunae), thus allowing the formation of preliminary thin trabeculae made up of woven bone that, due to their too-high cellularity, are not effectual from a mechanical viewpoint. Afterwards, along the surfaces of the SO-trabecular preliminary framework, dynamic osteogenesis occurs, which is mostly involved in filling primary haversian spaces, thus giving rise to primary osteons. DO-bone consists in lamellar bone which is mechanically more resistant compared to SO-trabecular bone, since it is less cellularized and arranged in a more orderly pattern; moreover, it occurs in relation to mechanical stimuli, instead of inductive vascular-derived factors (as occurs for SO).

On the contrary, if a blood supply deficiency occurs, thus leading to low local oxygen rate, cartilage may form within the fibrous tissue; eventually, the cartilage, after hypertrophy and calcification, will be replaced by bone, as in endochondral ossification. In the case of bone repair by endochondral ossification, SO never seems to take place. In fact, the osteoblasts in contact with the remnants of the calcified cartilage are directly arranged in movable laminae [52] and all appear to be functionally polarized in the same direction, i.e., toward the calcified cartilage. Thus, in endochondral ossification, DO is not preceded by SO.

At the end of the third stage, independently of the type of ossification (intramembranous or endochondral), the new bone that bridges the bone fragments is usually wider than the original bone profile. Once mechanical integrity has been re-established, the "remodeling stage" of the hard callus takes place. This represents the last stage of bone healing, which may lead to the recovery of the original anatomical shape. The balanced action of osteoclastic resorption and osteoblastic deposition is governed by Wolff's law and modulated by piezoelectricity, a phenomenon in which electrical polarity is created by pressure exerted in a crystalline environment [53].

## 5. Which Animal Model Is Suitable for Bone Investigation of Osteotomic Effects?

A great variety of animal models are present in the scientific literature (Table 2) for investigations of bone effects after osteotomy. As shown in Table 2, there are many similarities and differences regarding bone parameters concerning animal species, and between animals and humans, at the

same time. As if it were an aphorism, we can say that the animal model that most resembles human bone features is . . . the human being! In fact, no animal can perfectly mimic all the static and dynamic human bone characteristics [54]. Given the obvious ethical implications, studies of bone healing or bone damage on humans are very difficult. Nevertheless, investigations can be done by an indirect evaluation—for example, observing the biochemical values in the peri-implant sulcular fluid, the radiologic exams or certain protein (such as Hsp-70) expression in bone specimens collected immediately after an osteotomy [55,56].

**Table 2.** Main animal models for bone healing investigation [54,57–65].

| Animal | Bone Microscopic and Macroscopic Features | Bone Composition and Remodeling | Animal Management | Best Type of Study |
|---|---|---|---|---|
| Rodents | Mainly primary bone in long bone cortices and minimal cancellous bone. Cortices are thin and fragile. | Limited cortical remodeling and non-Haversian-type remodeling. Limited secondary osteon formation. Higher bone healing capacity in craniofacial bones. | Cheap and easily manageable. Rats are more docile and social than mice, although the latter are cheaper to house and maintain. | Osteoinduction. Cartilage regeneration potential. Bone infection. Extraoral surgical approaches. |
| Rabbit | Cortices are fragile and there is less cancellous bone than in humans. Quick achievement of skeletal maturity. Small size. Lack of biomechanical data. Dense Haversian bone. | Similar bone density to human. Bone metabolism is similar to human, with Haversian-type remodeling, although with a higher rate than humans. | Availability, housing and handling are easy. Cage confinement might worsen their bone healing capability. | Muscolo-skeletal research. Bone implants. Modeling of vertebral fracture repair. Extraoral surgical approaches. |
| Sheep | High trabecular bone density. Good body weight. Different bone microstructure than humans. Big difference between young and mature sheep due to age-dependent changes in bone structure. | Similar bone healing capability to humans. Different remodeling processes. | Although docile, their size requires a lot of space. | Orthopedic research. Bone filler materials in cranial osteotomies. Extraoral surgical approaches. |
| Goats | Good size. Presence of Haversian systems in the tibia, except for the caudal part. | Similar bone healing potential. Similar bone composition. | Docile and tolerant to environmental conditions. A lot of space is needed. | Bone filler materials in cranial osteotomies. Extraoral surgical approaches. Cartilage, ligaments and menisci regeneration. |
| Pig | Plexiform bone, which shifts to dense secondary osteonal bone. Good development of the Haversian system, with medium canals. Similar to humans. | Similar bone density and bone mineral concentration. Similar bone remodeling. | High body weight and aggressive nature. | Extra- and intraoral approaches. Osteonecrosis surgery. Osteogenic regeneration materials in craniofacial bones. Dental implants. |

**Table 2.** *Cont.*

| Animal | Bone Microscopic and Macroscopic Features | Bone Composition and Remodeling | Animal Management | Best Type of Study |
|---|---|---|---|---|
| Dog | Similar cancellous bone to humans. Presence of secondary osteons with small canals. Thinner articular cartilage. | Variability of trabecular bone remodeling depending on site, age and species. | Docile, easy handling. Good size. | Dental implants and peri-implantitis. |
| Non-human primates | Close to humans. | Comparable to humans. | Difficult to handle and highly trained staff are needed. | Bone implant. |

However, animal models are invaluable in the study of bone healing (Table 3). The European Commission stressed their central role and "the three Rs" have been established as the fundamental pillars of animal experimentation:

- Reduction: the number of animals involved must be the lowest necessary to achieve scientific evidence;
- Refinement: animals' suffering must be kept to a minimum;
- Replacement: if it is possible, a non-animal-based study is preferred [66,67].

**Table 3.** Advantages and disadvantages according to different times of euthanasia in animal models.

| | Advantages | Disadvantages |
|---|---|---|
| **Ex vivo** | Low costs<br>No need for ethical approval<br>Amount of information on bone damage<br>Reproducible conditions<br>Large numbers | Only hypotheses regarding possible clinical correlations<br>Less realistic setting |
| **In vivo and immediate sacrifice** | Evaluation of bone damage in an in vivo scenario<br>Possibility to obtain baseline results to compare with delayed sacrifice in in vivo results | Impossible to perform bone healing evaluations<br>Intermediate costs |
| **In vivo and delayed sacrifice** | Bone healing evaluation, with the possibility to obtain both static and dynamic evaluation of osteotomic gap<br>Wide range of parameters to be analyzed<br>Closer to clinical setting in humans | High costs<br>Small numbers<br>Small animal to respect animal testing hierarchy and ethics on animals |

These rules are imperative for an in vivo animal model. As far as the last point (replacement) is concerned, it has to be said that ex vivo studies (with bone segments provided by commercial slaughter) are suitable to evaluate osteotomic bone damage. Some authors also proposed artificial bone specimens derived from bovine bone due to its thermal conductivity, which is similar to human bone [68,69]. Altogether, a cost-benefit evaluation of the considered experimentation has to prove the potential for progress in human or animal health over the damage caused to the animals [54]. According to our experience, study-specific criteria must be considered as well. The aspects to take into consideration are:

- The type and the aim of the study
- The osteotomy device used.

The aim of the study has a central role in the correct choice of animal model: when the purpose is to study the osteointegration of endosseous dental implants, its size has to comply with animal and

bone-specific principles [59]. The same rule must be applied to the choice of osteotomy device: a large device must be used on adequately sized bones.

Together with these general principles, the choice of a proper animal model has to be influenced by other animal-specific criteria for bone:

- The specific bone site and its characteristics (femur, tibia, calvaria, mandible, vertebra)
- Bone macroscopic and microscopic features, including size and biomechanical properties
- Bone composition and density
- Bone turnover rates
- Specific conditions for bone healing, such as the critical size defect
- Costs and aspects regarding the management of the animal (including acquisition costs, housing costs and the need for specifically trained staff).

In particular, the economical and managerial aspects of animal experimentation have to be underlined. If a small animal model is possible, a larger animal model is to be considered unjustifiable according to Ethics in Animal Research Guidelines: (i) small animals before larger ones have to be selected to respect the animal testing hierarchy (European Communities Council Directive of 23 September 2010 [2010/63/EU]; (ii) compliance must be ensured with the Italian laws for the protection of animals used for scientific purposes (Decree Law No 61 of 4 March 2014) [70].

A particular animal can be highly suitable for a study because of its bone features, but there might not be a funding, logistic or training basis to proceed to the experimentation. At the same time, small animals such as rodents are often suitable in terms of feasibility, but they might not match the needed osseous characteristics, especially because of their small size.

Considering all the possible elements, the ideal animal models are non-human primates (NHPs), followed by dogs, although the ethical issues raised recently have made their involvement in animal experimentation increasingly difficult [71]. Nevertheless, it has to be considered that in vivo experimentation does not always imply the sacrifice of the animal—for example, in the case of an evaluation of microcirculatory parameters with intravital microscopy [72].

Bone macroscopic and microscopic characteristics deeply influence the choice of animal model: rodent bones can hardly be a suitable model for developing a novel dental implant, due to their small size. On the contrary, notwithstanding their dimensions, murine models can provide insights into the processes that underlie the biocompatibility of biomaterials [73–75] and bone healing, also based on multiple specific strains that have been selected over time [60,76].

Larger animals have many more clinical, biomechanical and dimensional similarities to humans. This is reflected also by the different bone turnover rates, which are close to the human ones in dogs or pigs and are much quicker in rodents and rabbits. Therefore, they can be crucial to study the steps and the clinical aspects of bone healing.

A question to be raised is whether a specific animal bone site (i.e., a specific region of a particular skeletal segment) is more suitable than another one in the study to obtain inferences about a specific human bone site. The most used sites in animal models are the femur, tibia, calvaria, mandible and vertebra, and each one has particular features that vary depending on the animal. A great variety of properties has to be considered in different subsites within the same bone, in term of bone density and cortico/cancellous thickness; for example, anterior and posterior mandible specimens show peculiar characteristics, as well as femoral diaphysis, metaphysis, epiphysis, et al. To our knowledge, there is no clear evidence in the scientific literature of the better adequacy of a specific anatomic site in an animal model for speculation about a human bone site. The investigator has to take into consideration all the above-mentioned aspects.

For instance, we chose as an animal model the in vivo rabbit calvaria in our previous paper [8], considering:

- The osteotomy instruments that we wanted to test (two different piezosurgical devices and one conventional rotary osteotomy device), which required high dense bone. We consequently excluded mice, dog alveolar bone and rabbit mandible or femur.
- The space that we needed in order to perform several osteotomic lines, which helped us in lowering the number of animals to sacrifice and led us to exclude mouse or rat as potential animal models due to their small size (Reduction in "the three Rs" [70]);
- The European Communities Council Directive, which led us to exclude large animal models (sheep, pork, dog) on an ethical basis [8].

## 6. Osteotomic Investigation with Rotational Instrument and Piezosurgical Devices: Our Experience

In 2018, we performed a study [8] on an animal model to evaluate, in 16 six-month-old white New Zealand rabbits, bone regeneration after osteotomy. The experimental procedures were executed in accordance with the Bioethical Committee of the Italian National Institute of Health and authorized with Decrees of the Italian Ministry of Health (protocol number 210/2013-B). Surgical procedures and animal care and maintenance were performed according to Italian law (D.L. no. 26/2014) and European legislation (EEC no. 63/2010).

The aim of the study was to compare bone healing dynamics in experimental osteotomies, using two piezosurgical devices with different output power (Piezosurgery® Medical and Piezosurgery® Plus) and a conventional rotary osteotome, in an in vivo rabbit model. The null hypothesis stated that there would be no difference in bone healing among the three devices employed.

Four couples of linear craniotomies (1 cm in length—full thickness cut of the calvaria preserving the dura mater) were carried out by the same surgeon in each rabbit skull, using different surgical osteotomic systems: (a) conventional rotary bur (RO); (b) Piezosurgery® Medical (PM) and (c) Piezosurgery® Plus (PP), both provided by Piezosurgery®—Mectron Medical Technology, (Carasco, Italy). PP is characterized by a higher output power (75 W) compared to PM (23 W) (Figure 1). The fascia-periosteal flaps and skin were eventually sutured. Two weeks after surgery, bone fragments were harvested and processed for histological and histomorphometric analyses as well as for scanning electron microscopy (SEM) investigations.

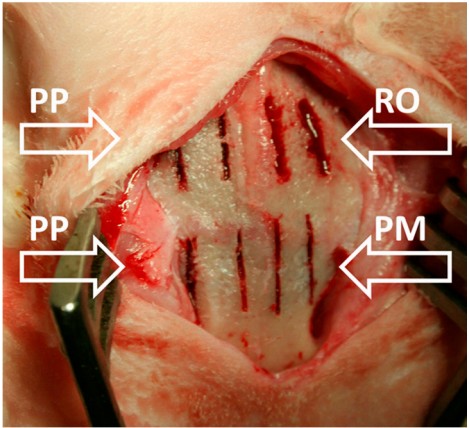

**Figure 1.** Gross photograph showing the osteotomies performed in each rabbit skull: two by means of rotary bur (RO), two by means of Piezosurgery® Medical (PM), and four by means of Piezosurgery® Plus (PP) (as indicated by arrows).

Results showed that osteotomies performed by means of both piezoelectric devices (PM and PP) produce half-sized bone gaps with respect to those produced by RO (Figure 2).

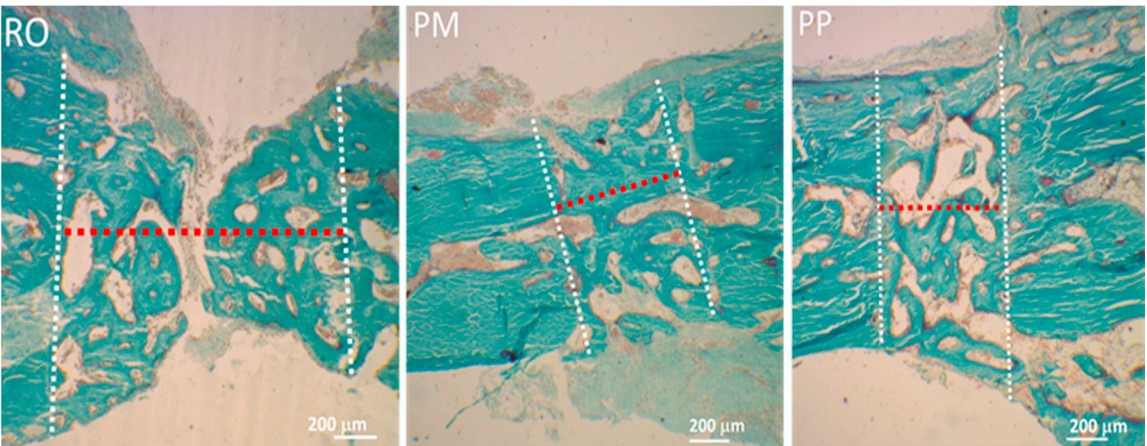

**Figure 2.** Light microscope (LM) micrographs showing the thickness (red dotted line) of the osteotomy (between the two white dotted lines) obtained using the three different devices (RO, PM, PP), producing different bone gaps. Note that the RO device produces a gap around twice the width of those produced by PM and PP. Perivascular stromal spaces among the bone-forming trabeculae appear wider in RO osteotomy.

Larger amounts of fibrous tissue (soft callus stage) with respect to bone tissue were present in RO samples than in PM and PP ones (Figure 3).

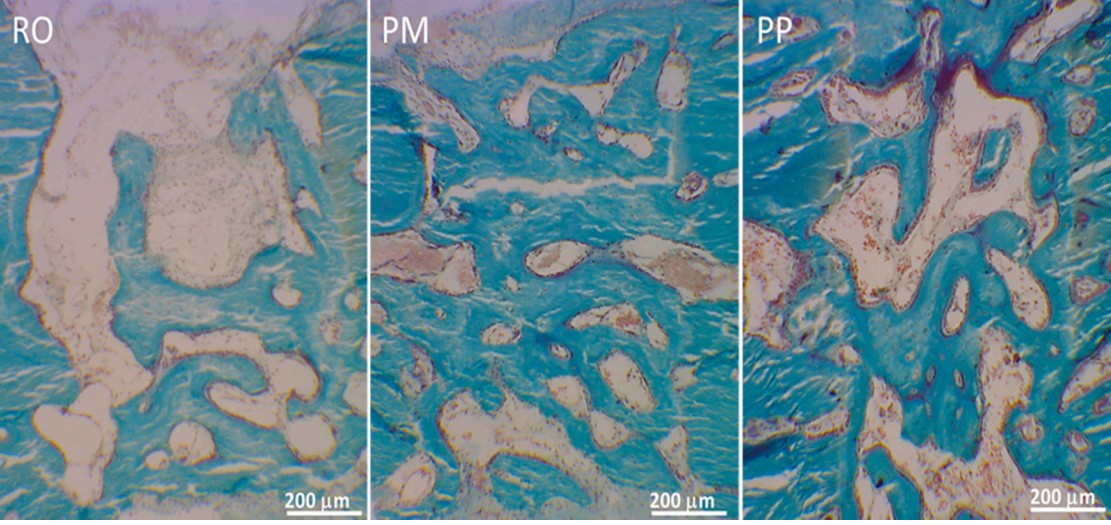

**Figure 3.** LM micrographs showing in three different osteotomies the newly formed bone during gap recovery.

Close to the forming bony trabeculae, numerous static osteoblasts arranged in cords and involved in preliminary bone regeneration were observed in RO samples; on the contrary, in PM and PP samples, bony trabeculae were mostly covered by typical prismatic dynamic osteoblasts arranged in monostratified laminae and involved in bone compaction (hard callus stage) (Figure 4).

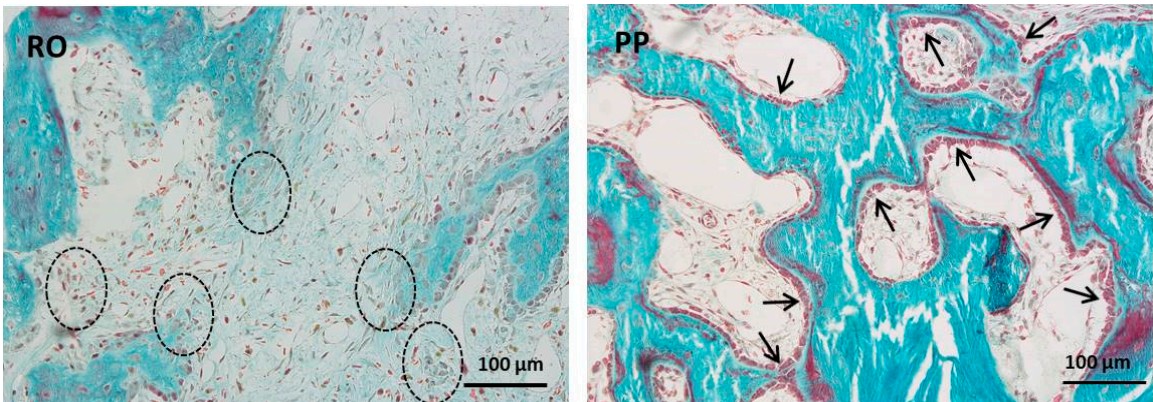

**Figure 4.** LM micrographs showing osteoblasts arranged in cords (dashed ovals) in RO. Arrows indicate osteoblastic laminae in PP field. It is also to be underlined that in the thinner PM/PP bone gaps, the osteoblasts are in more favorable conditions (i.e., a suitable distance from the capillaries) so that they more easily obtain the vascular derived inductive factors that, in turn, allow the progression from static to dynamic osteogenesis; as a consequence, a lower amount of fibrous tissue is observed in PM and PP with respect to RO.

Independently of the device used, in all osteotomies, backscattered SEM analysis showed, as expected, that the newly formed bony trabeculae inside the osteotomy gap were less mineralized with respect to the adjacent pre-existing bone; moreover, the regenerated bone was characterized by higher cell density with respect to the pre-existing bone. It is possible to observe inside the healing gap, only in RO osteotomies, the presence of some fragments/remnants of osteotomized bone, absent in PM and PP osteotomies (Figure 5).

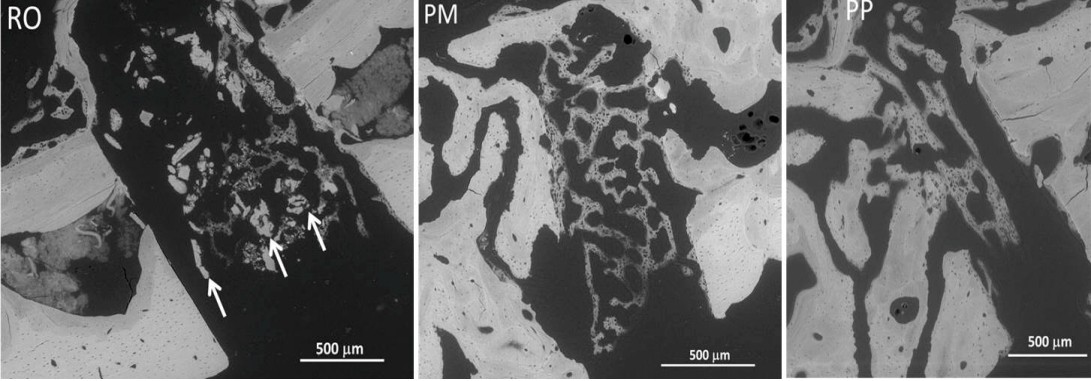

**Figure 5.** SEM micrographs showing in RO, PM, and PP samples the lower mineralization and higher cell density in the newly formed bony trabeculae inside the osteotomy gaps, compared to the adjacent pre-existing native bone. Some fragments/remnants of osteotomized bone are only present in RO osteotomies (arrows).

As far as bone regeneration is concerned (remodeling stage), the better performance of modern devices with respect to the traditional ones is also sustained by increased bone remodeling in PM and PP; in fact, a significantly higher osteoclast number (marked by tartrate-resistant acid phosphatase (TRAP) reaction) was observed within the gaps in PM and PP osteotomies with respect to RO ones, particularly concerning PP vs. RO (Figure 6).

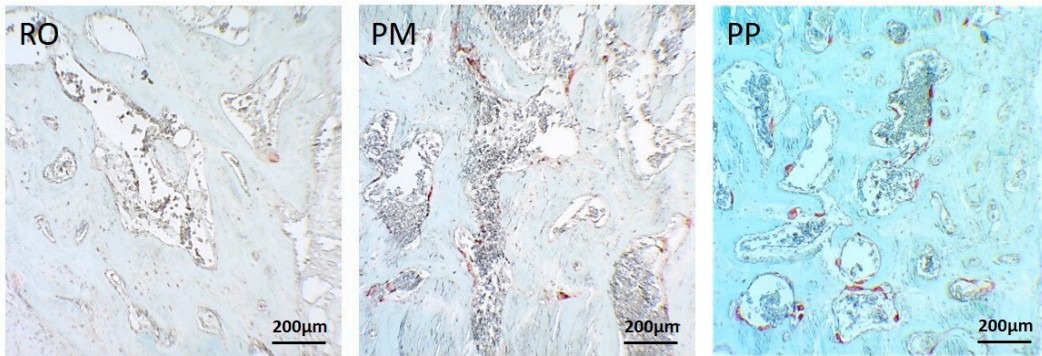

**Figure 6.** LM micrographs showing tartrate-resistant acid phosphatase (TRAP) reaction in the three osteotomies; in PM and PP osteotomies, the positivity (red color) is higher compared with RO.

To better explain, it is to be remembered that bone remodeling requires preliminary osteoclast differentiation/activation, in turn triggering the cells of the reversal phase (probably of stromal-fibroblast origin) that induce osteoblast differentiation, so that bone formation can occur to improve bone quality [77]. Besides the influence of an osteotomy device on bone regeneration, it is not of secondary importance its potential injury of bone cells near the edge of the osteotomy; our results show viable osteocytes closer to the osteotomy edge in bone cut by piezoelectric devices (in particular, concerning PP vs. PM) with respect to the traditional ones.

Finally, our findings demonstrate that PM and PP produce thinner bone gaps, in turn easier and faster to be recovered, with respect to RO osteotomies, where all healing processes require longer times. Therefore, piezosurgery is more effective than the conventional techniques in improving the progression of skeletal repair. According to these results, the null hypothesis that there would be no difference in bone healing among the three devices could be rejected.

## 7. What Kind of Information Can We Obtain from In Vivo and Ex Vivo Studies of Osteotomy?

Analyzing the scientific literature, great heterogeneity in the materials and methods utilized for studying osteotomic effects can be noted (Table 4).

**Table 4.** Main animal studies on osteotomic damage available in the scientific literature.

| Author | Animal Model | Type of Study | Material and Methods | Osteotomy Device | Conclusions |
|---|---|---|---|---|---|
| Moss, 1964 [78] | Dog mandible | Ex vivo | Ocular micrometer | Rotary cutting device set at three different speed ranges | Ultra-high-speed cutting did not produce higher bone damage than lower speed mode, especially when used with adequate coolants. Cooling agents reduced bone damage. |
| Spatz, 1965 [79] | Dog jaw | In vivo. The animals were sacrificed at 1, 2 and 7 days postoperatively | Photomicrography | Rotating burs at conventional and high speed. | Ultra-high-speed osteotomies showed better and quicker bone healing and smoother cut edge compared to conventional speed burs, particularly evident 1 week after the surgical procedure. |
| Horton et al., 1975. [80] | Dog alveolar bone | In vivo. The animals were sacrificed immediately after the surgical procedure and at 3, 7, 14, 28, 56 and 90 days postoperatively. | Light microscope | Ultrasonic instrument, low-speed rotary cutting bur and surgical chisel | The bur produced the smoothest surface. Healing in later periods appeared histologically to be the best with the use of the surgical chisel. After 90 days, the bone healing was complete. |

**Table 4.** *Cont.*

| Author | Animal Model | Type of Study | Material and Methods | Osteotomy Device | Conclusions |
|--------|--------------|---------------|----------------------|------------------|-------------|
| Eriksson et al., 1984 [81] | Rabbit femur, fibula and tibia | In vivo. Animals in the first group were sacrificed immediately after the surgical procedure. Half of the animals of the second group were sacrificed 1 week after the thermic damage, while the other half were followed up for 17 weeks. | Histology, histochemistry and vital microscopy. | Twist driller, thermostatically heated saline solution and thermal chamber | Histochemistry provides a better evaluation of bone viability after heat-induced trauma than histology. Vital microscopy is more sensitive in the evaluation of bone healing following heat trauma than indirect, histological and histochemical methods. |
| Wächter and Stoll, 1991 [82] | Bovine and sheep mandible | Both in vivo and ex vivo. The animals were not sacrificed. | Thermocouples | Oscillating saw | Only the combination of intermittent sawing and irrigation allows a safe osteotomy from a thermic point of view. |
| Abouzgia and Symington, 1996 [83] | Bovine femur | Ex vivo | K-type thermocouple (Omega) | Surgical drill (Stryker-100) | Drilling at high speed and large load seems to provide a better combination in terms of temperature rise and duration of temperature elevation. |
| Keijser et al., 1999 [84] | Rabbit tibia and femur | In vivo. The animals in the first group were sacrificed after 1 week. The animals in the second group were sacrificed after 1, 3, 5, 7, 9 and 12 weeks. | Histologic examination and temperature measurement. | Cryosurgery | No pathologic fractures were observed in rabbit tibiae. The gap in terms of periosteal bone apposition between human and animal bones was the likely cause of this difference. Rabbit bone, then, is not a suitable model to study this kind of biomechanical dynamics. |
| Bachus et al., 2000 [85] | Human femur | Ex vivo | Thermocouple | Standard surgical drilling | Higher load reduces the risk of thermal necrosis. |
| Vercellotti et al., 2005 [86] | Dog alveolar bone | In vivo. The animals were sacrificed immediately after the surgical procedure and at 14, 28 and 56 days postoperatively. Some bone specimens were collected immediately after the surgery as well. | Histomorphometric analysis | Piezosurgery Mectron Dental Technology, carbide bur and diamond bur | By day 56, the surgical sites treated by burs evidenced a loss of bone, versus a bone gain in the piezo-treated sites. |

**Table 4.** *Cont.*

| Author | Animal Model | Type of Study | Material and Methods | Osteotomy Device | Conclusions |
|---|---|---|---|---|---|
| Preti et al., 2007 [87] | Mini pig tibia | In vivo The animals were sacrificed at 7, 14, 28 and 56 days postoperatively. Some tibial bone specimens were collected immediately after the surgery as well. | Histomorphology and levels of bone morphogenetic protein (BMP)-4, transforming growth factor (TGF)-β2, tumor necrosis factor-alpha, and interleukin-1β and -10 were evaluated in the peri-implant osseous samples. | Piezosurgery Mectron Dental Technology and drilling according to Branemark protocol | Piezoelectric bone surgery appears to be more efficient in the first phases of bone healing; it induced an earlier increase in BMPs, controlled the inflammatory process better and stimulated bone remodeling as early as 56 days post-treatment. |
| Maurer et al., 2008 [88] | Rabbit skull | Ex vivo | Light microscopy, environmental surface electron microscopy (ESEM) and confocal laser scanning microscopy (CLSM). | Rotating instrument, micro-saw, Piezosurgery—Mectron Dental Technology | Bony structure integrity observed after the ultrasonic technique. |
| Queiroz et al., 2008 [89] | Rabbit tibia | In vivo The animals were sacrificed immediately after the surgical procedure. | Immunoistoc hemistry, scanning electron microscopy | Conventional drilling | It is preferable to use a less traumatic surgical protocol in order to preserve cell viability. After the thirtieth perforation, it is possible to observe a protein balance alteration. |
| Bertollo et al., 2010 [16] | Pig femur | Ex vivo | Infrared Thermal Imaging Camera (Digicam-IR, Ircon, Niles, IL, USA) | Pneumatic surgical handpiece, 2- and 3-fluted drills—7100 Drill MicroAire Surgical Instruments LLC, Charlottesville, VA, USA | The 3-fluted drills did not grant a clear advantage compared to 2-fluted drills in terms of better bone healing and screw fixation and of reduction of heat rise. |
| | Wether tibia | In vivo The animals were sacrificed 2 and 4 weeks after the surgical procedure. | Light microscopy— Olympus, Tokyo, Japan | Said surgical handpiece mounted in a sterilizable mobile drill-press. | |
| Rashad et al., 2011 [90] | Bovine rib | Ex vivo | Thermocouples | A conventional implant drill system (Straumann, Freiburg, Germany) and two ultrasonic osteotomic devices: Piezosurgery (Mectron Medical Technology) and Variosurg (NSK, Tochigi, Japan). | Implant site preparation with ultrasonic devices with adequate irrigation can provide an equally safe method compared to conventional drilling. |

**Table 4.** *Cont.*

| Author | Animal Model | Type of Study | Material and Methods | Osteotomy Device | Conclusions |
|---|---|---|---|---|---|
| Augustin et al., 2012 [91] | Pig femur | Ex vivo | Thermocouple | Combination of internally cooled drill and a two-step drill. | Internally cooled drill causes the lower temperature bone rise. |
| Heinemann et al., 2012 [92] | Pig jaw | Ex vivo | Light microscope. | Piezosurgery—Mectron Medical Technology, SONICflex and the conventional bur method. | The bone matrix adjacent to the defect radius showed intact osteocytes. |
| Hollstein et. al, 2012 [93] | Rabbit skull | Ex vivo | Light microscopy, (ESEM), and confocal laser scanning, microscopy (CLSM). | Piezosurgery 3, Piezon Master Surgery, Piezosurgery—Mectron Medical Technology, VarioSurg, Piezotome 2 | The osseous micro-structure is preserved. Five different piezosurgical devices were evaluated. |
| Schutz et al., 2012 [94] | Pig jaw | Ex vivo | Temperature sensors and digital volume tomography images. | Piezosurgery 3—Mectron Medical Technology | The correct use of the ultrasonic device allowed a safe osteotomy, not causing irreversible thermal damage in the bone. |
| Claire et al., 2013 [95] | Pig femur | Ex vivo | Scanning laser vibrometer, laser profilometer, scanning electron microscope | Piezosurgery 3 with an OP3 style insert tip—Mectron Medical Technology | In the cortical mode, the optimal load was of 150 g. The structure of the bone has to be taken into consideration as well. |
| Esteves et al., 2013 [96] | Rat tibia | In vivo The animals were sacrificed at 3, 7, 14, 30 and 60 days postoperatively. | Histomorphometric analysis, immunohistochemical staining, RT-PCR (reverse transcriptase-polymerase chain reaction). | Piezo Master Surgery and conventional drilling with a 2 mm round diamond coated tip. | Bone healing dynamics after piezosurgery are comparable to those observed with conventional drilling. |
| Gulnahar et al., 2013 [56] | Human mandible | In vivo Bone specimens were collected immediately following the surgical procedure | Heat shock protein 70 (Hsp70) expression. | Conventional bur and piezosurgery. | Conventional burs determined more aggressive procedures, showing significantly higher Hsp70 expression in consequence of the higher stress induced. |
| Ma et al., 2013 [97] | Rabbit skull | In vivo The animals were sacrificed at 1, 2, 3, and 5 weeks postoperatively. | Light microscopy, histomorphometric analysis. | Piezosurgery—Mectron Medical Technology, two types of oscillating steel saw blade. | Advanced bone healing compared to a traditional saw was observed. |
| Bullon et al., 2014 [98] | Bovine rib | Ex vivo | Thermocouple | Drilling with precipitation-hardening stainless steel (K drills) or with martensitic stainless steel (S drills). | Irrigation had a significant impact on heat generation, while drill use and type did not. |

**Table 4.** *Cont.*

| Author | Animal Model | Type of Study | Material and Methods | Osteotomy Device | Conclusions |
|---|---|---|---|---|---|
| Lamazza et al., 2014 [99] | Bovine rib | Ex vivo | Load cell and fluoroptic thermometer. | Piezosurgery with different tips (IM1s, IM2s, P2-3, IM3)—Mectron Medical Technology | Load, movements management and bone features play critical roles in temperature rise. Irrigation fluid temperature and the clogging effect also contribute to this phenomenon. |
| Stelzle et al., 2014 [100] | Pig skull | Ex vivo | Thermocouple, histomorphometric analysis. | Piezosurgery—Mectron Medical Technology, spiral burs and trephine burs | Piezosurgery generates more bone damage and higher temperatures than conventional drilling devices when used on high load levels. The maximum load should be 400 g. |
| Stoetzer et al., 2014 [72] | Rat calvaria | In vivo The animals were not sacrificed. The evaluations were performed immediately after the surgery and 3 and 8 days postoperatively. | Intravital microscopy of microcirculatory parameters. | Periosteal elevator and piezoelectric device. | A better periosteal microcirculation was found in piezoelectric device osteotomies. |
| Yang et al., 2014 [101] | Mice skull | In vivo The animals were not sacrificed. The evaluations were performed the day following the surgery and 2, 4 and 8 weeks postoperatively. | Micro-CT (micro–computed tomography). | Surgystar diamond round tip and Piezoelectric System (Synthes), | Piezosurgery provided faster bone healing in comparison with mechanical instrumentation. |
| Rashad et al., 2015 [102] | Bovine rib | Ex vivo | Thermocouple. | Conventional, sonic and ultrasonic osteotomic devices. | Sonic and ultrasonic devices provided the safer osteotomies in terms of heat generation. Irrigation was crucial to prevent temperature rise. |
| Tekdal et al., 2015 [55] | Human maxilla | In vivo The evaluations were performed until 24 weeks postoperatively, with a different timetable for each parameter considered. | Peri-implant sulcular fluid (PISF) analysis, periapical radiographs and cone beam computed tomography (CBCT). | Piezosurgery and conventional drilling. | On the biochemical side, piezosurgery provided a reduced inflammatory response of the bone, while on the radiographic analysis, conventional drilling and piezosurgery had similar crestal bone loss values. |
| Boa et al., 2016 [103] | Bovine rib | Ex vivo | Temperature measurement cavities. | Freehand drilling and surgical-guided drilling, combined with irrigation fluids at different temperatures. | Drilling through a surgical guide allowed the best results, especially in combination with the use of 10 °C pre-cooled irrigation fluid. |

**Table 4.** *Cont.*

| Author | Animal Model | Type of Study | Material and Methods | Osteotomy Device | Conclusions |
|---|---|---|---|---|---|
| Gabric et al., 2016 [104] | Rat tibia | In vivo The animals were sacrificed immediately after the surgical procedure and at 1, 2 and 3 weeks postoperatively. | 3D laser scanning technique (i.e., laser triangulation profilometry). | Piezosurgery—Mectron Medical Technology, Er:YAG laser both in contact mode and non-contact mode. | Osteotomies executed with Er:YAG laser in non-contact mode were the fastest to heal. |
| Lamazza et al., 2016 [105] | Bovine rib and femur | Ex vivo | Fiber optic thermometer. | Diamond tip—IM1s, Mectron Medical Technology. | Cortico-cancellous bone samples presented more variability in temperature values compared to cortical bone samples. |
| Sagheb et al. 2017 [106] | Porcine iliac crest and tibia | Ex vivo | Temperature measurement by infrared spectroscopy. Primary implant stability by resonance frequency analysis and extrusion torque. | IM1S; IM2P; IM3P; Piezosurgery—Mectron Medical Technology. | No significant difference in temperature increase between traditional drilling and piezosurgery. |
| Szalma et al., 2017 [107] | Pig mandible | Ex vivo | Thermocouple and infrared thermometer. | Diamond drills, tungsten carbide drills, piezoelectric diamond sphere and saw. | The use of irrigation fluid at 7 °C and pre-drilling is crucial to avoid a potentially nerve-damaging temperature rise in piezosurgical osteotomies. The speed of piezosurgery and of the other devices is similar. |
| Anesi et al., 2018 [8] | Rabbit skull | In vivo The animals were sacrificed 15 days following the surgical procedure. | Histology and enzymatic assay Histomorphometry. Scanning and transmission electron microscope. Nano-mechanical analysis | Piezosurgery Medical®, Piezosurgery Plus®, Mectron Medical Technology. Physiodispenser 7000, Nouvag AG. | Piezosurgical group shows more advanced stages of bone healing compared to conventional bur. |
| Favero et al., 2018 [108] | Sheep tibia | In vivo The animals were sacrificed at 1, 2 and 6 weeks postoperatively. | Eclipse Ci microscope (Nikon Corporation, Japan) with a digital video camera (Digital Sight DS-2Mv, Nikon Corporation, Japan). | Conventional drilling, both at high and mixed speed | There was no difference between the two groups. |
| Junior et al., 2018 [109] | Bovine femur | Ex vivo | T-type thermo-couple and Scanning Electron Microscope (SEM) | Piezoelectric tips (Driller) and tri-helicoid dental burs (Dentoflex) | The use of either rotatory burs or piezoelectric tips generates a temperature that does not affect the tissue healing. Burs create a smooth surface, and piezoelectric tips show a rougher and condensed bone surface. The wear of both systems does not cause a relevant increase in temperature after the preparation of 30 surgical beds. |

**Table 4.** *Cont.*

| Author | Animal Model | Type of Study | Material and Methods | Osteotomy Device | Conclusions |
|---|---|---|---|---|---|
| Lajolo et al., 2018 [110] | Porcine rib | Ex vivo | Implant site preparation with conventional drill system vs. piezoelectric system. Infrared thermometer was positioned underneath | Premium Surgical Kit Kohno, Sweden & Martina s.p.a. OP4 insert. Piezosurgery—Mectron Medical Technology | "Bone overheating using a piezosurgery unit is a potential risk during implant site preparation". |
| argued by Stacchi et al., 2018 [111] | Methodological flaws were revealed: OP4 insert is not suitable for implant suite preparation according to manufacturer's booklet. Excessive pressure load applied on the piezosurgical tip during implant site preparation. No correct movement of the piezosurgical handpiece was applied in the study. | | | | |
| Singh et al., 2018 [15] | Bovine bone | Ex vivo | Optical microscope | Vibrational bone drilling | Rotational speed is the major responsibility of heat generation, although all parameters considered affect the result. The best results were obtained with a mid-range rotational speed. |
| Stocchero et al., 2018 [112] | Sheep mandible | In vivo The animals were sacrificed at 5 and 10 weeks postoperatively. | Histomorphometric, μ-CT and biomechanical analysis | Two different drilling protocols | In the long term, no difference was observed between the two groups, although in the early period, there was greater cortical bone remodeling in the undersized preparation group. |
| Tepedino et al., 2018 [113] | Bovine bone | Ex vivo | Thermo-control laser, Scanning Electron Microscopy (SEM) and light microscope | Conventional rotating bur and Piezosurgery 3 with an OP5 insert—Mectron Medical Technology | Piezosurgery caused lower bone damage and temperature rise than the conventional rotating bur. |
| Zheng et al., 2018 [114] | Pig femur | Ex vivo | Infrared camera | Ultrasonically assisted drilling, conventional drilling | Ultrasonically assisted drilling provided less bone damage and generated lower temperatures. |
| Alam et al., 2019 [115] | Bovine femur | Ex vivo | Thermocouples, two-component dynamometer (Type 9271A, Kistler), system microscope (BX53, Olympus), digital microscope (DP22, Olympus) | Vibrational drilling—Orthofix, Italy | The best results were obtained by setting a lower drilling speed, a lower feed rate and a frequency of 20 kHz. |
| Pavone et al., 2019 [116] | Rat calvaria | In vivo The animals were sacrificed immediately after the surgical procedure and at 7, 15, 30 and 60 days postoperatively. | Histometric and histological analysis | Er,Cr:YSGG laser, trephine drill | The Er,Cr:YSSG laser osteotomy allowed the best healing in animals exposed to cigarette smoke. |
| Zhang et al., 2019 [117] | Tibetan pig femur and radius | In vivo | Light microscope | Drilling in five different bits geometries | Chisel edge, drill bit geometry, flute number, edges, steps and direction affect bone damage level and its characteristics. |
| Crovace et al., 2020 [118] | Dogs and cats; calvaria or spine | In vivo. Ostetotomic bone immediate sampling; the animals were not sacrificed. | X-ray or CT scan. Histology | Piezosurgery, Mectron Medical Technology | No signs of bone overheating. |

Most of the studies investigated the effects of the heat generated by the metal–bone interface during drilling: indeed, the resulting friction can cause thermal osteonecrosis. Thermal damaged bone is of great clinical relevance because it is not capable of holding the implants and screws for a long time [15].

Well documented experimental osteotomic studies were conducted in the late 1950s, when Thompson [119] reported the "index of viability", i.e., a parameter for evaluating the distance between the first normal osteocyte and the edge of the osteotomy; it indicated the drill's cutting effects upon the cellular elements of the bone. This index can be carried out on ex vivo and in vivo samples and this author well distinguished for the first time between ex vivo and in vivo studies on animals [119]. A zone of aseptic thermal necrosis was described by Thompson in 1958 [119], which was characterized by the degeneration of the osteocytes (pyknosis of the nuclei, complete disintegration of the cells, osteocyte lacunae devoid of cellular elements). Decreased bone damage is indicated by a low viability index value and larger bone damage by the higher values.

In the mid-1960s, it was demonstrated that increased bur cutting speed with a cooling agent does not cause increased bone damage, i.e., decreased bone damage occurred if increased speed was applied [78]. Ultraspeed or increased speed was also supported by in vivo findings of Spatz [79], in terms of lesser inflammatory response and faster bone recovery.

A rudimentary ultrasonic device was verified in the 1970s, found to be related to slower bone healing compared to rotational bur [80]. Ultrasonic and piezoelectric osteotomy was developed and rediscovered in the early 2000s; on the contrary, bone healing outcome proved to be faster with piezosurgical devices with respect to a traditional rotational bur [8].

In 1983, Eriksson et al. investigated the friction and the heat generated from the metal–bone interface during drilling: it was eventually established that the lowest temperature threshold for thermal osteonecrosis is 47 °C for one minute. This study was particularly informative because it was conducted in an in vivo animal model (rabbit) and represents nowadays a pillar of the relevant literature [120].

Conflicting conclusions in terms of the effect of temperature in bone tissue can be seen by comparing in vivo and in vitro studies, as Wächter et al. [82] concluded in their study. The scholar revealed lower bone temperature in in vivo samples: bleeding flow could evacuate more thermal energy with respect to ex vivo settings [82].

Post-osteotomy callus formation and bone healing are influenced by the biological status and potency of the bone at the cut surfaces [121]. Excessive heat and thermal necrosis may irreversibly impair the bone healing process [122]. To limit the temperature increase, both bur and oscillating saws are usually cooled during the cutting process [123].

Thermal damage in the surrounding cortical bone depends not only on the maximum temperature value but also on the duration of temperature elevation and consequently on drilling time. High speed and increased force in drilling cause a small rise in temperature, due to the decrease in the drilling time [83].

Drilling with forces between 57 and 130 N is related to a minor increase in temperature, and similarly, the time interval in which the temperature remains above 50 °C is shorter [85]. Moreover, the results from Bachus contradict the work published by Abouzgia et al. [85] which had even shown that the cortical bone temperatures decreased between 1.5 and 9.0 N.

Factors that influence heat generation during bur osteotomy were investigated by several studies and they include speed, drill force, irrigation, drill design and drill diameter. Another widely used technology as a standard tool for osteotomies is the oscillating saw [102].

From Vercellotti's [86] paper to nowadays, most studies have compared the effects on bone of traditional rotary osteotomes versus ultrasonic (piezosurgical) devices. The research interest in ultrasonic osteotomy is justified by the fact that piezosurgical osteotomy is extremely precise and provides arbitrary cut geometries, prompt handling, efficient bone cutting and minimal damage to adjacent soft tissue structures [102].

Vercellotti et al. [86] applied an in vivo model of dog alveolar ridges, demonstrating an improved bone healing process in the piezosurgically treated group. The index of viability of bone cells after piezosurgery was studied histologically by numerous authors in bone grafting procedures for

dental purposes [124,125]. Histomorphometric and molecular analysis was evaluated together by Preti et al. [87]; the authors established that piezosurgery may accelerate the earlier phases of the dental implant osteointegration when compared with rotational drilling.

Bone surfaces were examined by Maurer [88] with three microscopic techniques (light microscopy, environmental scanning electron microscope—ESEM, confocal laser scanning microscope—CLSM) after using different procedures of osteotomy (reciprocate micro-saw, Lindemann bur, ultrasonic osteotome). ESEM and CLSM entail an appraisal of unmodified bony specimens, being non-destructive examination techniques. In this histological investigation, ultrasonic osteotome activity was not related to osteonecrosis of the trabecular bone. CLSM permits a quantitative evaluation of bone surface roughness [88].

Augustin and co-workers [91] in an ex vivo model studied in depth the thermal damage to bone as a combined result of the temperature and the duration of elevated temperature: the relation of maximum bone temperature and the period of that increased temperature over the critical value was described. Drilling with carbide spiral drills (2.5–4.5 mm) for dental implantology, the bone temperature persisted around 50 °C for 50 s in 95% of results, not adequate for avoiding thermal osteonecrosis during drilling (≤47 °C for one minute, as demonstrated by Eriksson et al.) [120].

Comparing different ex vivo inserts for piezosurgical osteotome, it was found that cortical bone thickness had surprisingly no influence on intraosseous temperature generation, which remained, however, below the threshold of 47.8 °C for 1 min, without irreversible thermal damage in the bone [94]. These data were statistically supported and in contrast to the previous study of Eriksson, which was conducted with traditional rotary osteotomes [81]. Considering the investigation methods, these results are difficult to compare. The author concluded that in vivo measurements need to be carried out to confirm the data.

While cortical bone thickness appeared a secondary factor following piezosurgery, the structure of the bone seemed to be an important factor in the cut characteristics: higher cut lengths and widths were present in the spongeous bone compared to the cortical bone [95].

Esteves et al. [96] conducted an in vivo study on rat tibia (histological, histomorphometric, immunohistochemical and molecular analysis); comparative appraisal of bone recovery after osteotomic lines performed by rotating drill or piezosurgery did not demonstrate significant differences between the two animal groups.

An in vivo histomorphometric study by Ma et al. [97] revealed no statistically significant differences in comparing bone healing after osteotomy performed with piezosurgery versus oscillatory saws; increased bone remodeling activity in early phases was observed for the piezoelectric surgery group.

Cortico/cancellous samples (ex vivo) are a simulation of in vivo conditions frequently used by scholars, but they are affected by greater variability of results. In an ex vivo bone model with a piezosurgical device, three main factors were finally identified among the others for having a great influence on the heat production: working load, working movement management and bone features [99]. Rashad et al. in 2015 found that ultrasonic osteotomies cause significantly lower heat compared to rotary bur osteotomies [102]. However, both these scholars concluded that (in vivo) animal studies were essential for understanding how bone healing occurred after ultrasonic osteotomies and how blood flow and biological factors fit together with ex vivo findings [99,102].

In a rotary bur (drilling bur), it was found by histopathology that the heat generation is mainly dependent on rotational speed: at high rotational speed (3000 r/min), there is a severe thermal osteonecrosis [15]. These results contradict previous findings in the 1960s and 1970s, when ultraspeed or high rotational speed were recommended for avoiding thermal osteonecrosis [78–80].

Modulating the irrigation temperature can be an effective strategy in reducing heat during osteotomy with a drilling bur. Studies on endosseous dental implant site demonstrated that the use of precooled irrigation at 10 °C entails a significant reduction in peak temperature compared with what was achieved when irrigating at room temperature [103].

Bone removal approximating the inferior alveolar nerve canal was investigated through an ex vivo experimental study conducted by Szalma [107]. Compared to a drill, this ex vivo study

confirmed that piezoelectric bone removal approximating the inferior alveolar canal resulted in higher temperature-increasing effects on the surface of the inferior alveolar nerve [107]. The author eventually suggested combined bone removal (i.e., tungsten carbide pre-drilling completed with piezoelectric tip) with 7 °C pre-cooled irrigation to limit the intracanal temperature increase [107].

During endosseous dental implant site preparation, bone overheating using a piezosurgery was measured, resulting in two times more with respect to a conventional drill [110].

Augmented bone overheating with piezosurgical devices compared to a bur drill was an unexpected finding. Other authors questioned some of the methods that led to these results, such as Stacchi et al. [111], who discussed the previous ex vivo investigation of Lajolo et al [110]. For example, ultrasonic tip choice in the experiment can lead to abnormal heat generation if the use is improper for the purpose (side-cutting insert instead of dental implant site preparation insert). Stacchi et al. (2018) also described that applied pressure load on the handpiece was excessive for that device and this could induce a higher temperature rise [111]. Pressure load should be chosen according to the specific characteristics of the tip and device (piezosurgical, rotatory, and oscillating or reciprocating saw). Moreover, as in the paper of Lajolo et al [126], the laboratory supported arm for the handpiece should reproduce the movement of the hand in the operative theater (sliding movement is not comparable to rotating movement). Non-overlapping conditions between experimental and clinical setting could lead to distorted interpretation of results from studies conducted in laboratory settings [111].

Previous findings of Lajolo [126] and Szalma [107] about bone overheating with piezosurgical devices were questioned by Zheng [114] in a further ex vivo model. He found that the cortical bone temperature during piezosurgical osteotomy was lower than during rotational drilling. Moreover, according to this study, the impact of the drilling factors on bone temperature was in the order of drill diameter, rotational speed, feed speed and the frequency and amplitude of the piezosurgical vibration [114].

Erbium-doped yttrium aluminum garnet (Er:YAG) laser was applied as osteotome in maxillo-facial surgery in 2007 [10]. A single study in vivo compared a Er:YAG laser with the piezosurgical technique, without histomorphometric evaluation but with bone volume measurement by laser triangulation profilometry [104]; considering the limit of investigation methods, the Er:YAG laser showed faster bone healing with respect to the piezosurgical sample. Pavone et al. (2018) [116] carried out an in vivo study in rats exposed to inhalation of cigarette smoke, comparing a Er,Cr:YSGG laser (erbium, chromium: yttrium, scandium, gallium and garnet) and trephine drilling. The Er,Cr:YSGG-S group revealed higher bone formation with statistical significance compared to the trephine drilling group.

Microcracks are mechanical complications that can be considered a further parameter to evaluate bone damage after the osteotomy procedure and fixation/prosthesis positioning [127,128]. If the gap formed by the microcracks around the prosthesis surface overtakes 50 mm, the capacity of the bone tissue growing into the prosthesis will be decreased [127,128], proving that drill geometries influence the mechanical and thermal damage: a standard surgical drill edge (chisel) seems to augment bone damage.

## 8. Future Perspectives

Several ex vivo studies on animal bone were recently dedicated to heat production during osteotomy with conventional drills and piezoelectric devices, particularly in endosseous dental implant sites. In our opinion, outcomes from ex vivo animal models cannot be transferred to clinical practice, because the static finding of bone damage does not necessarily correlate with the effectiveness of the bone healing process (Table 3). Discussing ex vivo results, scholars should only advance perspectives as regards the in vivo conditions, both on animal models and in surgery, since it is not known whether the described bone damage has consequences for bone healing.

The informativeness of the measured data depends on the time of execution, i.e., in tissue out of living (ex vivo) or in living tissue of an animal (in vivo). In turn, in in vivo experiments, histology can be carried out at different stages: immediately after cutting procedures, during the bone healing process and at the end of bone restoration.

Micro-mechanics, histology, ultrastructural analysis, molecular biology and biochemistry, both in ex vivo or in vivo, are altogether necessary to elucidate the sophisticated processes leading to bone restoration after the cutting procedure.

Therefore, all the above-mentioned techniques, integrating and completing each other, should contribute to shedding new light on the very complex issue of bone healing after osteotomy.

**Author Contributions:** Conceptualization, A.A. and M.D.B.; validation, L.C.; writing—original draft preparation, M.D.B., A.P. and R.N.; writing—review and editing, A.A., F.C. and M.F.; visualization, R.S.; supervision, C.P. All authors have read and agreed to the published version of the manuscript.

**Funding:** This research received no external funding.

**Conflicts of Interest:** The authors declare no conflict of interest.

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
