# Peer review of "Bone Healing Evaluation Following Different Osteotomic Techniques in Animal Models: A Suitable Method for Clinical Insights"

_applsci, doi:10.3390/app10207165_

Round 1

Reviewer 1 Report

This is an interesting article with a lot of invaluable scientific information that could be well applied in clinical practice.

I've got only 2 minor suggestions to the authors:

  1. Miss spelling of "Cytotology" at Page 4, Table 1.
  2. In Page 8, line 308. The authors shared their own experimental procedures on rabbit calvaria using conventional rotary burs and 2 different Piezo devices. I think all readers would also be very curious about how deep those craniotomy cuts were, not just the length described in the manuscript.

Author Response

Manuscript ID: applsci-950452. Authors’ Answers to Review Report Form

We thank the reviewer for the appropriate and useful observation, which deserve specific answers.

Reviewer 1

Comments and Suggestions for Authors

This is an interesting article with a lot of invaluable scientific information that could be well applied in clinical practice.

I've got only 2 minor suggestions to the authors:

  1. Miss spelling of "Cytotology" at Page 4, Table 1.

The spelling of “Cytology” was corrected.

  1. In Page 8, line 308. The authors shared their own experimental procedures on rabbit calvaria using conventional rotary burs and 2 different Piezo devices. I think all readers would also be very curious about how deep those craniotomy cuts were, not just the length described in the manuscript.

We agree with the reviewer. The sentence (now in page 8, line 312) was rewritten as follows. “1cm in length - full thickness cut of the calvaria preserving the dura mater”.

  1. English language and style are fine/minor spell check require

Numerous and widespread corrections of English language in the text have been carried out. These are highlighted with the notes to the text.

Reviewer 2 Report

The authors provide a narrative review entitled Bone Healing Evaluation Following Different Osteotomic Techniques in Animal Models: A Suitable Method for Clinical Insights.
The review covers some fundamental aspects of the matter with proper narrative methods and appropriate language. A large number of articles support this review. However, this reviewer suggests updating some literature about piezoelectric surgery. Some other recently published paper has been published in animal surgery and, in my humble opinion, they are worthy to be considered in this review. (i.e. 10.3390/vetsci7020068).
I suggest the editor consider the current manuscript for publication after minor changes.

Specific question.
line 366 - the figure is improperly positioned, cropped and not cited in the text. Please revise.

Author Response

Manuscript ID: applsci-950452. Authors’ Answers to Review Report Form

We thank the reviewer for the appropriate and useful observation, which deserve specific answers.

Reviewer 2

Comments and Suggestions for Authors

The authors provide a narrative review entitled Bone Healing Evaluation Following Different Osteotomic Techniques in Animal Models: A Suitable Method for Clinical Insights.

The review covers some fundamental aspects of the matter with proper narrative methods and appropriate language. A large number of articles support this review. However, this reviewer suggests updating some literature about piezoelectric surgery. Some other recently published paper has been published in animal surgery and, in my humble opinion, they are worthy to be considered in this review. (i.e. 10.3390/vetsci7020068). We agree with the reviewer. We have entered the paper 10.3390/vetsci7020068 in Table 4.

I suggest the editor consider the current manuscript for publication after minor changes.

Specific question.

line 366 - the figure is improperly positioned, cropped and not cited in the text. Please revise. We agree with the reviewer.  Fig.6 was replaced.